# Optical Coherence Tomography for 3D Weld Seam Localization in Absorber-Free Laser Transmission Welding

**Frederik Maiwald [1,*]**, **Clemens Roider [2]**, **Michael Schmidt [2,3]** and **Stefan Hierl [1]**

1 Laser Material Processing Laboratory, Ostbayerische Technische Hochschule Regensburg, Am Campus 1, 92331 Parsberg, Germany; stefan.hierl@oth-regensburg.de
2 Institute of Photonic Technologies, Friedrich-Alexander-Universität Erlangen-Nürnberg, Konrad-Zuse-Straße 3/5, 91052 Erlangen, Germany; clemens.roider@lpt.uni-erlangen.de (C.R.); michael.schmidt@fau.de (M.S.)
3 Erlangen Graduate School in Advanced Optical Technologies (SAOT), Paul-Gordan-Straße 6, 91052 Erlangen, Germany
* Correspondence: frederik.maiwald@oth-regensburg.de; Tel.: +49-9492-8384-108

**Featured Application: Optical coherence tomography enables the three-dimensional inspection of internal structures like the weld seam in polymer parts and has excellent prerequisites for process monitoring in the optical and medical industries.**

**Abstract:** Quality and reliability are of the utmost importance for manufacturing in the optical and medical industries. Absorber-free laser transmission welding enables the precise joining of identical polymers without additives or adhesives and is well-suited to meet the demands of the aforementioned industries. To attain sufficient absorption of laser energy without absorbent additives, thulium fiber lasers, which emit in the polymers' intrinsic absorption spectrum, are used. Focusing the laser beam with a high numerical aperture provides significant intensity gradients inside the workpiece and enables selective fusing of the internal joining zone without affecting the surface of the device. Because seam size and position are crucial, the high-quality requirements demand internal weld seam monitoring. In this work, we propose a novel method to determine weld seam location and size using optical coherence tomography. Changes in optical material properties because of melting and re-solidification during welding allow for weld seam differentiation from the injection-molded base material. Automatic processing of the optical coherence tomography data enables the identification and measurement of the weld seam geometry. The results from our technique are consistent with microscopic images of microtome sections and demonstrate that weld seam localization in polyamide 6 is possible with an accuracy better than a tenth of a millimeter.

**Keywords:** laser transmission welding; transparent polymers; optical coherence tomography; process monitoring; image processing

## 1. Introduction

Numerous medical and optical devices are made of transparent polymers. The manufacturing of these devices often takes place in cleanrooms and places high demands on cleanliness, precision, visual appearance, and reliability. Absorber-free laser transmission welding possesses several advantages (i.e., contactless input of energy, high precision, no adhesives and no particle formation) which enable the fulfilment of the aforementioned demands [1–3]. In contrast to conventional transparent-absorbent welding, where one partner contains a laser absorbent additive [2,4,5], or an additional absorbing material is placed in the joining zone [6–9], the samples analyzed in this work are made of identical polymers and were joined directly using absorber-free laser transmission welding [10–16]. Figure 1 shows the process principle of absorber-free laser transmission welding. The setup is comparable to the one described by Olowinsky [17]. Both joining partners are

clamped in an overlap. A laser beam with the wavelength in the polymers' intrinsic absorption spectrum between 1.6 μm and 2 μm is used. Focusing the beam with a comparably high numerical aperture (NA) provides significant intensity gradients inside the specimen, which enables the selective fusing of the joining zone [1,13,14]. The joining of two 1 mm thick partners with a 0.05 mm to 0.6 mm wide weld seam, for example, is possible without affecting the surface of the upper joining partner [11,14].

The main challenge in absorber-free welding is the precise generation of the weld seam in the horizontal and vertical direction along the desired trajectory. If the seam extends too far vertically to the upper surface, a visible and tactile bulge occurs. An overly short seam insufficiently penetrating both joining partners causes instability and inadequate tightness [1,10,12]. Nevertheless, horizontal alignment is also important, since cells or drugs can temporarily linger and lump in remaining gaps if a fluidic channel is not sealed exactly at its edge. This creates an unacceptable risk to the patient using such a device and requires process control [12,18].

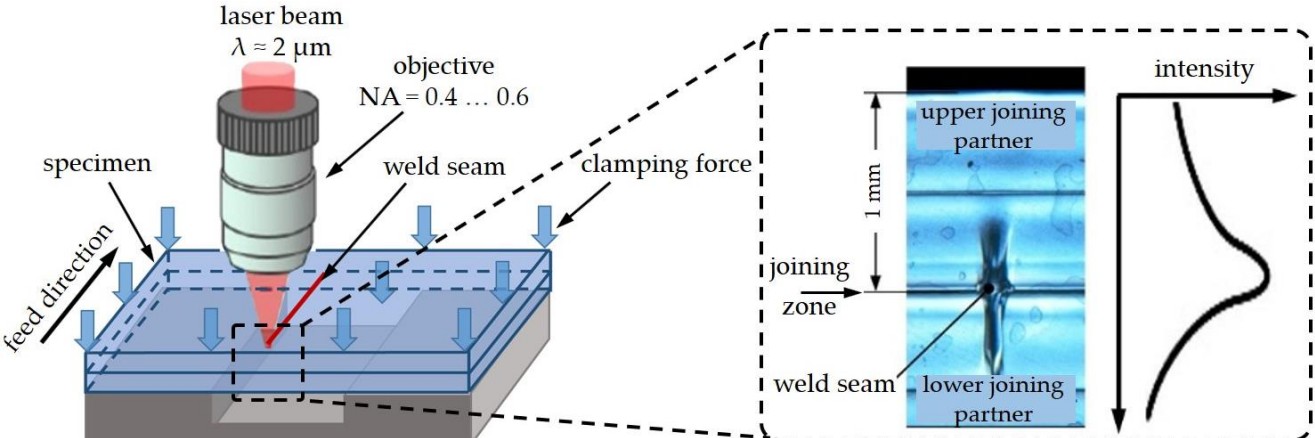

**Figure 1.** Sketch of the experimental setup for absorber-free laser transmission welding (**left**) and an exemplary weld seam photographed in polarized light (**right**) (adapted from [19]).

In addition to integral tests of the complete product (i.e., burst pressure, leakage and drop-down tests) [4,20], a separate weld seam analysis is necessary to validate the welding process. Conventionally, thin sections of the weld seam area are prepared by grinding or cutting the specimen with a microtome. Extracted sections are viewed in a transmitted light microscope with polarized light. The weld seam becomes visible in the 2D image due to the change in optical properties (i.e., birefringence) during plasticization and solidification (Figure 1, right) [5].

A three-dimensional (volumetric) examination is possible via computer tomography (CT), enabling the detection of geometric errors (i.e., gaps and bubbles) and melt blowouts into fluidic channels by quantifying the density change between the polymer and combustion gases or surrounding air. Unfortunately, the density difference of the base and the weld seam material is usually too small for weld seam detection via CT. A major disadvantage of the methods discussed above is that they are useful for detecting errors only in post-processing.

In contrast, pyrometry enables online (in-situ) temperature monitoring during processing. Deviations can be identified and compensated using feedback between temperature and processing parameters. Using this closed-loop control, overheating in corners and bubble formation can be prevented [12]. Furthermore, the temperature signal is linked to the vertical expansion of the weld seam, enabling indirect weld seam localization in the vertical direction [12,15]. However, misplacement of the seam or overheating cannot be clearly distinguished from one another as they both superimpose in the pyrometer signal, and no direct information about the weld seam geometry is given by pyrometry [18].

In conclusion, the usability of absorber-free laser transmission welding is still limited, since the possibilities for process monitoring are still insufficient. We aim to improve this issue by introducing optical coherence tomography for 3D weld seam localization.

Optical coherence tomography (OCT) is a technique that uses surface reflections and scattered light within a sample to generate 3D images of a geometry [21]. To enable OCT, we employ a spectral-domain OCT (SD-OCT) attached to a 2D scanner (Figure 2). In the OCT, the low coherence light of a superluminescent diode is divided into a sample and a reference path of an interferometric setup. The reflected light from the sample is collected and coupled back into the fiber. After combining it with the light from the reference path, the resulting interferogram is measured with a spectrometer. As the modulation of the interferogram contains the depth information of the sample, the depth-reflectivity profile $R_{(z)}$ is extracted using a Fourier transformation of the output data [21]. By scanning in two directions, the one-dimensional data (A-scan, in the $z$-direction) is expanded volumetrically to obtain a three-dimensional reflectivity profile $R_{(x, y, z)}$ (C-scan). Optical coherence tomography has excellent prerequisites for online monitoring of laser-based processes: data points can be acquired with several hundred kilohertz, and a scanner is usually already there for processing.

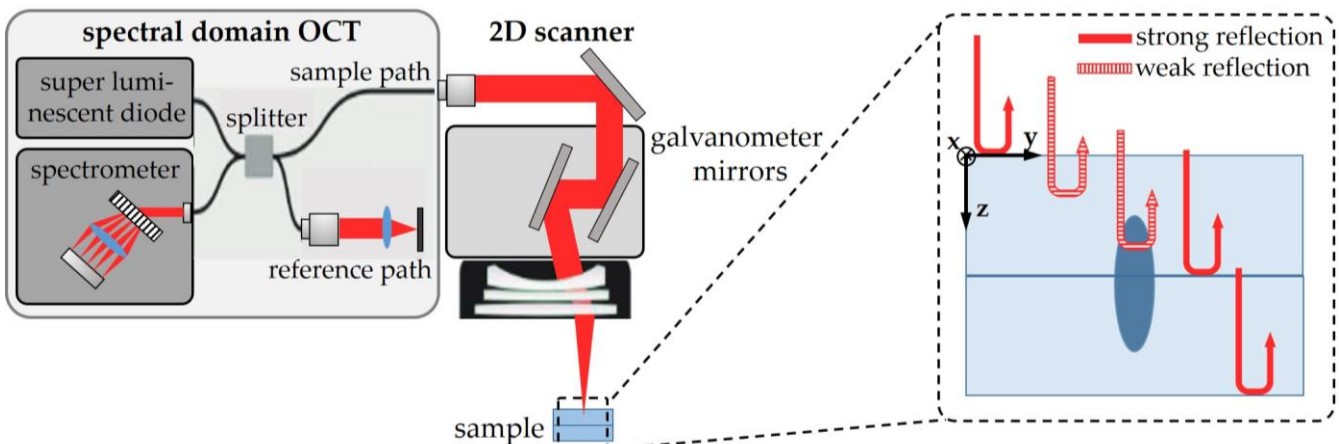

**Figure 2.** Sketch of the scanner-integrated SD-OCT for volumetric weld seam analysis (adapted from [22]).

OCT has already been used for non-destructive testing in different industrial applications of laser transmission welding. Usually, interfaces between different parts or materials are determined based on the abrupt change in optical properties. Since the upper joining material is partially transparent, a highly reflective surface for welding metal sheets to plastic [23] and internal structures with high differences in the refractive index like glass fibres or gas bubbles can be recognized [24–27]. In transparent-absorbent laser transmission welding, OCT has already been used for measuring the seam width and for detecting gaps and internal pores [22,28–31]. If there are no errors, the interface between a properly welded transparent and absorbent partner is usually represented by an absent OCT signal [31]. However, because the absorbing partner contains additives, changing the optical properties compared to the transparent one, the interface of properly welded partners can be detected using an OCT with sufficient dynamic range [29].

Ultra-short pulsed laser welding of glass is similar to absorber-free laser transmission welding of polymers, as samples of an identical material or with very similar optical properties are joined. Weld seams in glass are localized via OCT by detecting gas-filled cavities [32] at the top and bottom of the seam [33]. In glass welding, therefore, there is an abrupt change in the refractive index at the interface from glass to gas.

In conventional transparent-absorbent transmission welding as well as glass welding, there is always a boundary between different materials (polymer to gas, polymer to metal,

transparent to absorbent polymer, glass to gas), which leads to strong reflections induced by abrupt changes in optical properties. In contrast, the material of both joining partners is identical in absorber-free laser transmission welding. Therefore, volumetrically resolved transitions within the polymer—from the injection-molded base material to regions re-melted during welding—are resolved to obtain the weld seam dimensions in this work. Gross defects, such as gas bubbles due to thermal decomposition, which would result in an easily detectable interface, are unlikely if the process is carried out appropriately and are not part of this study.

## 2. Materials and Methods

### 2.1. Experimental Setup for Laser Transmission Welding

For welding, a thulium fiber laser ($\lambda$ = 1940 nm, TLR-120-WC-Y12, IPG Laser GmbH, Burbach, Germany) with 120 W (cw) power was used. Figure 1 shows the experimental setup. The fixed-focus objective with an adjustable, comparably high NA of 0.4 to 0.6 was mounted on an optical rail with slides. The NA was set to a value resulting in a Rayleigh length of 0.3 mm inside the material. A fine-threaded spindle moved the rail, enabling the variation of the distance between the optics and the specimen and, thus, the adjustment of the laser focus position. A measurement system with 0.01 mm resolution controlled the rail's position. A clamping device with a conical slit hole fixed the 2 specimens ($50 \times 15 \times 1.05$ mm$^3$ each) in an overlapping position. It is moved by a 2-axis linear system enabling feed rates $v$ of up to 300 mm/s. We have already presented a detailed description of the experimental setup, the processing parameters and the simulation-based process layout [14]. The samples are made of semi-crystalline polyamide 6 (Ultramid B3S) provided by BASF SE (Ludwigshafen am Rhein, Germany) and molded by Gerresheimer Regensburg GmbH (Regensburg, Germany) with a coefficient of absorption $\alpha$ of 0.8 1/mm at a wavelength of 1940 nm.

The welds were processed at 3 different laser focus positions, $z_{focus}$, of 1.0 mm, 1.1 mm, and 1.2 mm at a 200 mm/s feed rate between 0.14 J/mm and 0.46 J/mm energy per unit length ($E$). The value of $z_{focus}$ = 1.0 mm was set to the joining zone by processing and evaluating little seams. For every parameter setting, one sample containing at least six weld seams was manufactured.

### 2.2. Conventional Weld Seam Analysis Using Microtome Sections

To measure the weld seam geometry manually, approximately 50-$\mu$m thick cross-sections of at least 7 welds per parameter setting were prepared using a rotary micro-tome (Leica RM2255, Leica Microsystems Ltd., Shanghai, China). The sections were photographed in polarized light using a transmitted light microscope (Olympus BX53M, Olympus Deutschland GmbH, Hamburg, Germany). Afterward, the weld seam areas were measured manually using the image processing software ImageJ and OLYMPUS Stream Essentials Version 2.3.

### 2.3. OCT Data Acquisition and Image Processing

For weld seam analysis with OCT, a Telesto II OCT (Thorlabs GmbH, Bergkirchen, Germany) with an acquisition rate of 76 kHz and a central wavelength of 1.3 $\mu$m was used with the ThorImage 5.3.1.0 software in combination with the LSM02 objective (Thorlabs), giving a depth or so-called axial ($z$) resolution of 5.5 $\mu$m and a lateral resolution of 7 $\mu$m ($1/e^2$ beam diameter in $xy$). For each weld seam, a volume (C-scan) of $5.0 \times 1.5 \times 2.4$ mm$^3$ ($x, y, z$), consisting of $400 \times 500 \times 1024$ data points, was recorded within 18 s. The distance in the $x$-direction between two B-scans ($yz$-planes) was 12.5 $\mu$m. This is more than 1.75 times larger than the lateral resolution of the OCT, and the B-scans are therefore independent of each other and do not share any information. Within a B-scan, the pixel sizes of 3 $\mu$m in the horizontal ($y$) and 2.3 $\mu$m in the depth ($z$) direction are below the resolution of the OCT with the used objective as recommended by Thorlabs. The dispersion compensation

was quadratic with a dispersion factor of 63. The A-scan averaging was set to 5, and the refractive index of the base material was assumed to be 1.5.

Figure 3a shows an example of a volumetric view of the recorded data in the ThorImage OCT software. After the acquisition, the data were transferred directly to ImageJ using the interface implemented in ThorImage. The subsequent image processing was performed automatically using self-written scripts. In ImageJ, the data were converted to 8-bit and saved as a TIFF image stack first. Figure 3b shows the first B-scan of the stack, and Figure 3c shows the final processed B-scan, including a sketch of the weld seam measurements. After processing all B-scans, they were concatenated into a 3D volume (Figure 3d).

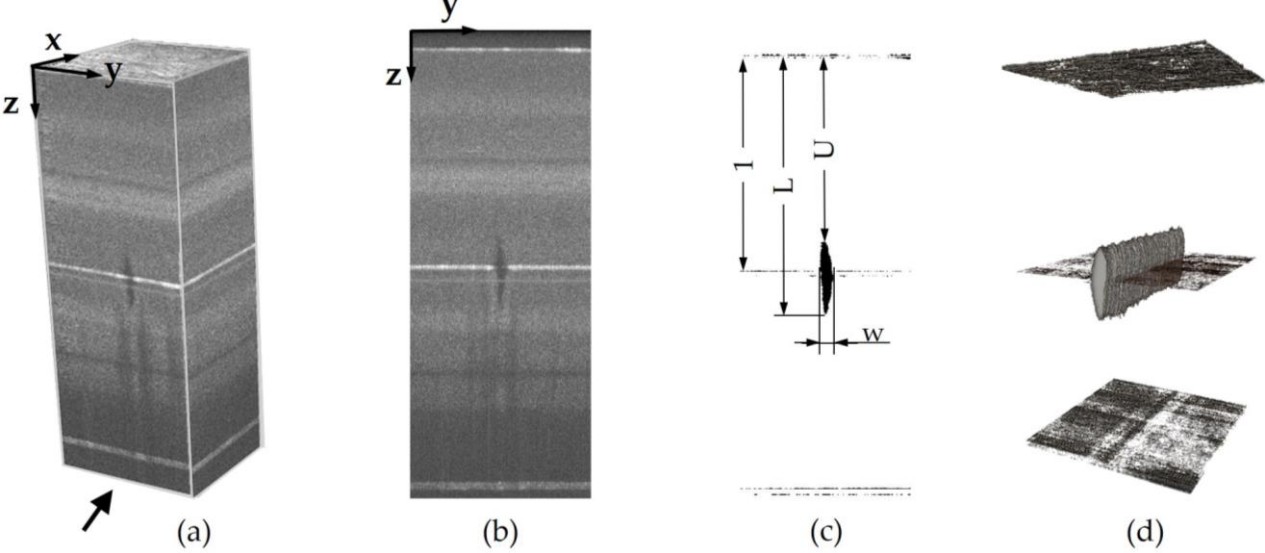

**Figure 3.** (**a**) Volumetric view (C-scan) of the unprocessed data; (**b**) *yz*-slice (B-scan) of the data; (**c**) processed image of the slice with the weld seam measurements; (**d**) volumetric view of the completely processed data. The view shows the margins of the joining partners, the joining zone in the middle and the adjacent weld seam. Base area: $0.7 \times 0.7$ mm$^2$. Material: Ultramid B3S. Processing parameters: Rayleigh length = 0.3 mm, $\lambda$ = 1940 nm, $\alpha$ = 0.8 1/mm, $v$ = 200 mm/s, $E$ = 0.22 J/mm.

During image processing, the margins of the upper and the lower joining partner and the joining zone between both partners were determined first. Secondly, the weld seam and its dimensions were identified. Both steps were performed with separate scripts. Details of the image processing are explained below.

### 2.3.1. Determination of the Sample Margins

The margins of both joining partners were determined for every B-scan by simple thresholding since they are already clearly visible as bright lines in the raw data. After improving the contrast using the "*enhance contrast*" function with default parameters (leading to a mean greyscale of $138.7 \pm 3.5$ and a standard deviation of $53.8 \pm 1.2$ of the histogram), the image was converted to binary, where greyscale values between 230 and 255 were set to white (255) and all others were to black (0). After removing the remaining noise using the "*remove outliers*" function with a block size of 10 pixels in the horizontal direction and one pixel in the vertical direction (chosen because the margins are horizontal as well), and one pixel standard deviation, the obtained binary image contained the margins of both the upper and the lower joining partner, and the data were transferred as CSV to MATLAB. In MATLAB, the centroid of the pixels of the upper surface was calculated for every B-scan. After removing outliers, a 2nd-degree polynomial was fitted, since the C-scan is slightly curved due to a lack of image field correction and the sample might be bent. The value of the polynomial function was used as the starting point for measuring the weld seam within a B-scan.

### 2.3.2. Determination of the Weld Seam

Scattering and absorption within the sample attenuate the detected OCT signal with depth ($z$-direction). In deeper regions, the measurement beam is weaker, and a smaller portion of reflected light reaches the detector. This leads to a signal/brightness gradient in the data, which makes automated evaluation difficult. To overcome this issue, the average brightness of every image row ($y$-direction) was calculated in MATLAB. The result was smoothed using a moving mean over 50 rows and was saved as a greyscale image with constant brightness per row, which was used as a background image. Thereafter, the original image was divided pixel wise by this background image using the "*Calculator Plus*" function in ImageJ. This gave an image in which the detected signal was independent of the depth-dependent attenuation.

To determine the area of the weld seam, the contrast was improved using the "*enhance contrast*" function with default parameters and the image was converted into binary. In contrast to the previously mentioned identification of the margins, the "*isodata*"-algorithm [34] and no fixed threshold were used. Since the weld seam does not always clearly differ from the surrounding bulk material, several regions (possible weld seam areas) remained in the B-scan. To remove noise and refine the selection, after using "*remove outliers*" with radius 5 and threshold 50, the blockwise "*remove outliers*" function was utilized, where the horizontal ($y = 30$) and the vertical ($z = 5$) size of the rectangle used for filtering can be defined. This makes use of the fact that in a B-scan, the longer axis of the weld seam is vertical ($z$-direction), enabling the distinction from horizontally ($y$-direction)-aligned features caused, for example, by the melt flow during injection molding.

After using the "*find connected regions*"-algorithm, which only detects areas larger than 1500 pixels (0.01 mm$^2$), the remaining areas were filtered by size and position. All regions adjacent to the left, right or bottom margin of the scan, larger than 20,000 pixels (0.11 mm$^2$) or not containing any pixel in the region 0.12 mm above or below the joining zone were not transferred to MATLAB. In MATLAB, the width $w$ and the distances between the upper surface and the upper ($U$) and lower ($L$) end of the weld seam were measured, and the seam height $h$ ($L - U$) was calculated. Subsequently, the aspect ratio $k$ of width to height was checked ($k = h/w$). Only areas with an aspect ratio between 3 and 5 were considered, and remaining outliers (defined as a value more than three scaled median absolute deviations away from the median) were removed. The image was concatenated into a three-dimensional array, and the next image was processed. Figure 3d shows the result after processing all images. All data were recorded and processed with identical parameters. We determined the mentioned parameters once for the best identification of U and L in a few B-scans, and no further adaptions, e.g., depending on the weld seam size, were performed.

## 3. Results and Experimental Verification

Figure 4 shows the distance between the specimen's surface and upper ($U$) and lower ($L$) end of the seam (a), as well as the seam width $w$ (b). The results obtained with the OCT are colored, and the reference measurements obtained with thin cuts are black.

Considering the reference values (black), the distance between the specimen's surface and the upper end of the seam $U$ varies between 0.44 mm ($E = 0.35$ J/mm, $z_{focus} = 1.0$ mm) and 0.99 mm ($E = 0.30$ J/mm, $z_{focus} = 1.2$ mm) and the distance between surface and lower end L varies between 1.12 mm ($E = 0.14$ J/mm, $z_{focus} = 1.0$ mm) and 1.54 mm ($E = 0.46$ J/mm $z_{focus} = 1.2$ mm). The seam width $w$ varies from 0.05 mm ($E = 0.16$ J/mm, $z_{focus} = 1.1$ mm) to 0.16 mm ($E = 0.43$ J/mm, $z_{focus} = 1.2$ mm). Shifting the focus and, therefore, the seam downward in the material decreases the seam height, whereas the width is nearly unaffected. Both seam height and width increase with increasing laser power. Considering the OCT measurements (colored), the weld seams with the highest and lowest energy are missing (except for $z_{focus} = 1.1$ mm). The processing of the missing weld seams is, in principle, possible but would require other image processing parameters. With other parameters, however, other seams would not be recognized.

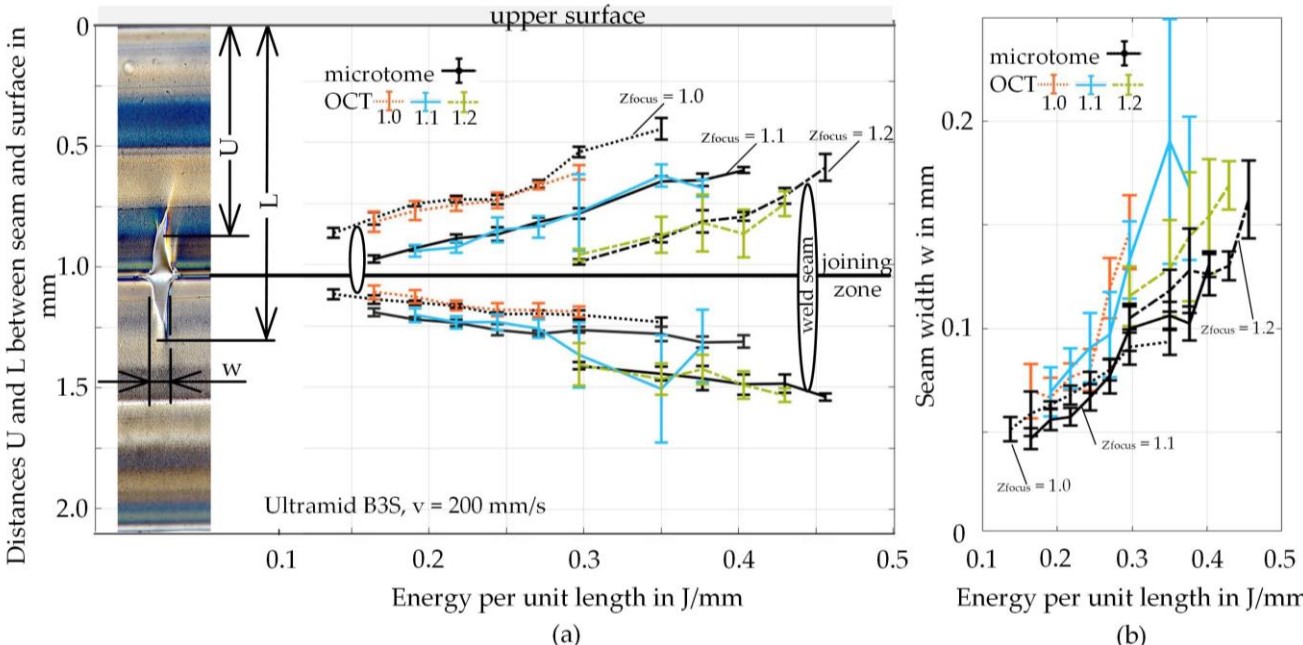

**Figure 4.** (**a**) Distance between specimens' surface and upper (*U*) and lower (*L*) end of the seam as well as seam width *w* (**b**) in dependence of energy per unit length (*E*) at a 200 mm/s feed rate. Material: Ultramid B3S. Rayleigh length = 0.3 mm, $\lambda$ = 1940 nm, $\alpha$ = 0.8 1/mm. At least six welds per parameter setting were analyzed. The microtome data were obtained from [14].

The maximum distances *U* and *L* at which OCT measurements were possible are 0.96 mm (*E* = 0.30 J/mm, $z_{focus}$ = 1.2 mm) and 1.53 mm (*E* = 0.43 J/mm, $z_{focus}$ = 1.2 mm). The average deviation between OCT and the reference measurement for the distance *U* is 0.025 ± 0.021 mm, and the maximum is 0.08 mm (*E* = 0.30 J/mm, $z_{focus}$ = 1.0 mm). For the distance *L*, the average deviation is 0.034 ± 0.05 mm and the maximum is 0.23 mm (*E* = 0.35 J/mm, $z_{focus}$ = 1.1 mm). The height (*L* − *U*) of detectable seams is 0.26 mm (*E* = 0.19 J/mm, $z_{focus}$ = 1.1 mm) to 0.87 mm (*E* = 0.35 J/mm, $z_{focus}$ = 1.1 mm), and the width is 0.07 (*E* = 0.19 J/mm, $z_{focus}$ = 1.1 mm) to 0.19 mm (*E* = 0.35 J/mm, $z_{focus}$ = 1.1 mm). However, since this is the seam with the largest deviation by far, we also mention the second-largest seam with 0.78 mm height and 0.17 mm width (*E* = 0.43 J/mm, $z_{focus}$ = 1.2 mm).

The average deviation of the weld seam width *w* is 0.027 ± 0.022 mm, and the maximum is 0.084 mm (*E* = 0.35 J/mm, $z_{focus}$ = 1.2 mm). The seam width obtained with the OCT is always larger than the reference one. Our explanation for this is the melt squeeze out into the gap between the joining partners. This squeeze out was only evaluated as part of the weld seam in the OCT, but not in the thin cut measurements. It is possible to evaluate only the width without the melt blowout from the OCT data by, e.g., applying a fit to the detected contour. We intentionally included the squeezed material, as it will contribute to the weld seam's strength and can disturb the fluid channels of the manufactured part.

The detection of the weld seam is not possible in every B-scan. Figure 5 shows the average distance in the feed direction (*x*) between two B-scans with successfully detected weld seams. The average distance between two consecutively detected weld seams is 0.04 mm, and the largest distance is 0.08 mm. Since the distance between two initially recorded B-scans is 0.0125 mm, this means that the automated weld seam identification successfully identifies seams in approximately 17% to 33% of B-scans.

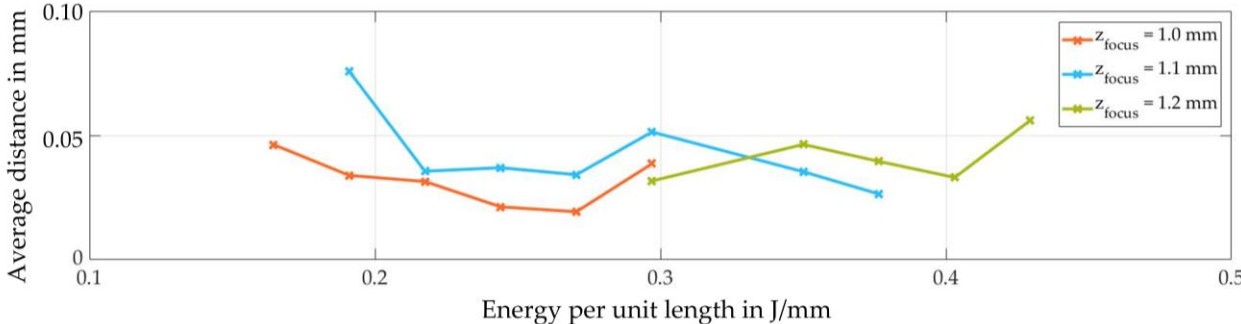

**Figure 5.** Average distance between two consecutively detected weld seams in the feed direction with B-scans taken every 0.0125 mm.

## 4. Discussion

The presented measurement method offers seam localization of welds from 0.25 mm to 0.95 mm in height and 0.07 mm to 0.17 mm in width with an accuracy of 0.03 mm compared to microtome sections. The results achieved are equally good regarding accuracy and probability of weld seam identification across the successfully analyzed welding parameter fields. All data were recorded and processed with identical OCT and image processing parameters. This shows that the proposed method works well for typical weld seams in absorber-free laser welding without the adaptation required to the weld seam geometry. The use of a different OCT system or the examination of other materials would require manual adjustment of the image processing and measurement parameters. The method has so far only been successfully tested on semi-crystalline PA6.

Insufficient contrast to surrounding dark areas is the main issue for unsuccessful weld seam detection. Figure 6 shows OCT data after background correction and contrast enhancement (left sides) and the reconstructed weld seams (right sides) before filtering regarding size and position. Particles at the surface lead to black streaks superimposing the weld seam (Figure 6a). The main problem, however, is the dark regions that appear in the lower joining part (Figure 6a,b). Bordering dark areas of the base material are no longer detected separately but added to the area of the seam. The weld seams (a) and (b) are sorted out by the filtering procedure, as they show unrealistic, abrupt changes and are several times larger than the average ones. In examples (b) and (c), there are only 2 frames' distance between the rejected seam (b) and the problem-free detection (c).

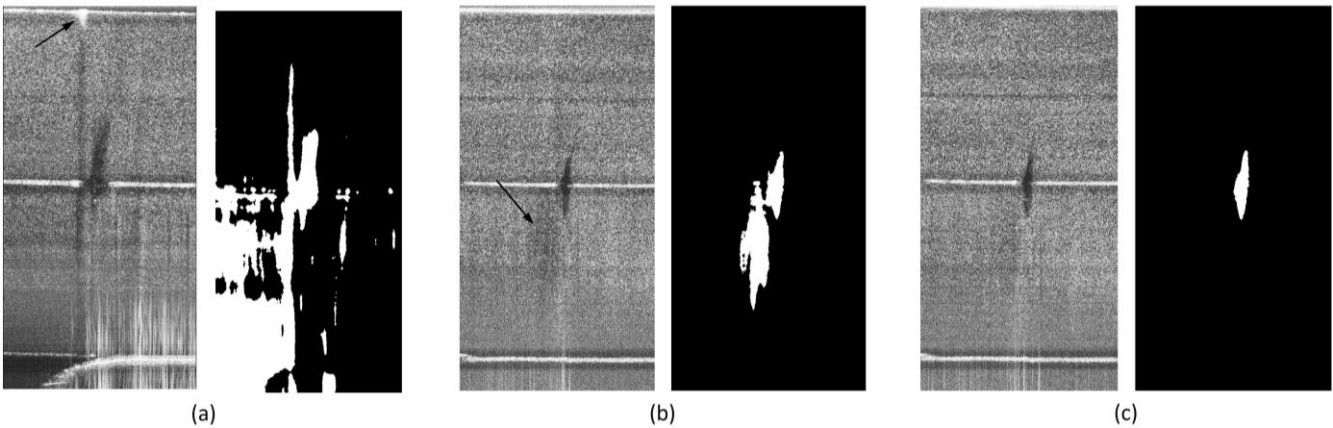

**Figure 6.** OCT data after background correction and contrast enhancement (left sides) as well as the reconstructed seams (right sides). $E = 0.35$ J/mm, $z_{focus} = 1.1$ mm, B-scan 248 on seam number 4 (**a**). $E = 0.24$ J/mm, $z_{focus} = 1.1$ mm, B-scan 4 (**b**) and B-scan 7 (**c**) on seam number 7.

The sample with the greatest overall deviation ($E = 0.35$ J/mm, $z_{focus} = 1.1$ mm) has a comparably large number of dark areas, and it was unfavorably labeled with a black pen in the area of the weld seams. Initially, we related the dark areas to our marking of the samples. They often occur below the marking but also occur independently of it. However, due to the specified cleanliness in industrial applications, particles and markings are a minor problem, and the detection rate will probably increase.

Even if weld seam identification is successful in only 16% to 33% of B-scans, the average distance between detected seams is less than 0.1 mm in our case, with a spacing of 0.007 mm between consecutive B-scans. This is far more than is practical with microtome sections for post-process evaluation. For potential in-process monitoring, our current setup allows for approximately 200 B-scans per second and, therefore, about 50 successfully identified seams per second. We believe this to be sufficient in most applications, and a faster acquisition is possible by reducing the lateral field-of-view (the horizontal scan range) or sacrificing lateral resolution.

We assume that the weld seam is clearly visible in PA6 because of changes in morphology during melting and solidification, even if the morphology of the weld seam has not yet been analyzed. This hypothesis is based on the findings by Hierzenberger that changes in morphology can be detected using a similar (not polarization-sensitive) OCT during extrusion [35]. Since the weld seam is clearly visible in polarized transmitted light (Figure 1 right and Figure 4a left), the use of a polarization-sensitive OCT should improve detection and might enable it with amorphous materials as well, since birefringence caused by stress becomes visible [26,36–38]. However, polarization-sensitive OCT is much more complex, which might limit its industrial use.

## 5. Conclusions

We showed that optical coherence tomography can be successfully used for three-dimensional weld seam localization in absorber-free laser transmission welding of polyamide 6. This method enables the automatized, three-dimensional evaluation of weld seam size and position inside the volume of welded samples. Compared to the state of the art of OCT in laser transmission welding, where the image contrast is due to boundaries of different materials or from a transition from polymer to gas, in our method, the weld seam is recognized by the morphology change caused by welding. It is therefore applicable to joining identical materials. The results obtained coincide well with data obtained by microtome cuts. The deviation of the weld seam size compared to thin cuts with polarized light is 0.03 mm, and the image processing algorithm works in a broad range of different seams from 0.26 mm to 0.87 mm in height and 0.45 mm to 0.96 mm distance from the surface. The algorithm identifies seams in about 25% of the measured B-scans.

The proposed method is not only robust but also sufficiently precise for weld seam monitoring and seems well suited for quality assurance in absorber-free laser transmission welding. Due to the OCT's excellent prerequisites, studies with different materials and online process monitoring are scheduled. To provide real-time feedback, the algorithm would need to be implemented on a graphics card. Incorporating further feature detection algorithms might increase the amount of successfully identified weld seams.

**Author Contributions:** Conceptualization, F.M.; data curation, F.M. and C.R.; formal analysis, F.M.; funding acquisition, M.S. and S.H.; investigation, F.M. and C.R.; project administration, F.M. and S.H.; resources, M.S. and S.H.; software, F.M.; supervision, M.S. and S.H.; validation, F.M.; visualization, F.M.; writing—original draft, F.M.; writing—review & editing, C.R., M.S. and S.H. All authors have read and agreed to the published version of the manuscript.

**Funding:** This research was funded by the Bavarian Ministry for Economic Affairs, Media, Energy and Technology, grant number DIE0113.

**Institutional Review Board Statement:** Not applicable.

**Informed Consent Statement:** Not applicable.

**Data Availability Statement:** Due to the large size of the raw data set, which was several hundredths of a gigabyte, the dataset will be provided on reasonable request.

**Acknowledgments:** The authors gratefully thank the project partner Arges (Novanta Europe GmbH) for their good teamwork. Thanks to Gerresheimer Regensburg GmbH for providing samples and to Futonics Laser GmbH for providing a laser. We personally thank our colleagues J. Tröger and E. Escher for their creative and helpful advice regarding OCT and image processing, as well as our colleague A. Dzafic for proofreading.

**Conflicts of Interest:** The authors declare no conflict of interest. The funders had no role in the design of the study; in the collection, analyses, or interpretation of data; in the writing of the manuscript, or in the decision to publish the results.

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
