# Peer review of "Optical Coherence Tomography for 3D Weld Seam Localization in Absorber-Free Laser Transmission Welding"

_applsci, doi:10.3390/app12052718_

Round 1

Reviewer 1 Report

This paper attempts to develop a method to determine weld seam location and size for selective fusing of laser beam inside the work piece internal joining zone without affecting the surface of the device. However, the paper needs improvement before it can be published in the journal.

Below points to consider for improvement:

  • The introduction and background are not clear.
  • Literature review is poor.
  • It seems an incremental work of author’s previous work. However, every manuscript should be a standalone paper and thus experimental details should be clearly described.
  • Detailed scientific analysis should be provided for the experimental results.
  • Discussions should be separated from the conclusion.
  • Conclusion should highlight the clear findings of the results probably with numerical values.
  • What is the limitation of the current study? How it can be overcome?

Author Response

We thank the reviewers for their detailed comments. We made corresponding changes which have significantly improved the manuscript. Below we list detail responses to each question/concern in blue.

Reviewer 1:

This paper attempts to develop a method to determine weld seam location and size for selective fusing of laser beam inside the work piece internal joining zone without affecting the surface of the device. However, the paper needs improvement before it can be published in the journal.

Below points to consider for improvement:

  • The introduction and background are not clear.
  • Literature review is poor.

In the section form line 104 to 130, we added information on research on using an OCT in laser transmission welding with an absorbing lower joining partner. We have detailed, why the method works in this well-established configuration (the contrast in the OCT image is due to the difference in optical properties of the two materials) and why this is not directly applicable for the welding of identical (transparent) polymers we study in this work. To the best of our knowledge there is no literature on OCT in absorber-free laser welding. We hope that his addresses the second comment by the reviewer as well as addresses the first one.

  • It seems an incremental work of author’s previous work. However, every manuscript should be a standalone paper and thus experimental details should be clearly described.

In this manuscript we focused on the automated detection of weld seams with an OCT. While the samples are identified by the used welding parameters, our intention is not to qualify the welding process or make any assertions on the laser processing. We address the new method to analyse already welded samples. Therefore, we did not include any detailed description of manufacturing of the samples as we believe this would shift/obscure the focus of the manuscript. In our opinion, it is also not an incremental work of ours as we did not publish anything on the OCT based evaluation of weld seam geometries for absorber-free transmission welding in a peer-reviewed journal. However, we presented a work-in-progress of this manuscript at the conference Lasers in Manufacturing, which should be no problem according to the webpage of Applied Sciences. While there are several publications on laser-transmission welding of polymers with added, in this work we use a different contrast mechanism in OCT. Thus, we think that this manuscript is a standalone paper as well as it is no incremental work.

  • Detailed scientific analysis should be provided for the experimental results.
  • Discussions should be separated from the conclusion.

We agree with the reviewer that a separate section discussion should be provided. We did so (see section 4 in the revised version of the manuscript) and hope that with the added discussion and rework of the section results, we satisfied the reviewer’s demand for a detailed scientific analysis.

  • Conclusion should highlight the clear findings of the results probably with numerical values.

With the suggested separation, the conclusions are now more focussed on the main findings/highlights of the manuscript and we added numerical values to support the claims.

  • What is the limitation of the current study? How it can be overcome?

These are valid questions which help improving the manuscript, although the second one is not so easy to answer.  We've added the new Discussion chapter as recommended to answer these questions.

One limitation is that our method was only tested for PA6 and we attribute the contrast we see in the OCT images to changes in the morphology in the weld seam due to melting and re-solidification. This might not work on amorphous materials, but a polarisation sensitive OCT could be still able to detect the weld seam as it is sensitive to birefringence.

Another limitation currently is the speed of the algorithm, which was not of our concern at this proof-of-concept stage. For real-time feedback / in-process detection, the algorithm would probably need to be transferred to a graphics card to increase the speed. Although we think that the current percentage of successful identification will be sufficient in most cases, it might be improved by incorporating advanced feature detection capabilities into the algorithm.

As we see that as an outlook, we added the following in the conclusions:

“For providing real-time feedback, the algorithm would need to be implemented on a graphics card and incorporating further feature detection algorithm might increase the amount of successfully identified weld seams.”

Although the automated evaluation without adaption of parameters works over a quite large range of weld seam sizes, too small or too large seams were not detected as mentioned in the manuscript. We believe that this is not of concern as this corresponds to a very large deviation from the desired weld seam (geometry). Nonetheless, the previously mentioned improvements to the algorithm might also increase the detectable size range.

Reviewer 2 Report

The paper is well written and the method used to obtain the dimensions of the seam extensively described.

Nevertheless, some clarifications could help improving the paper:

- Lines 162-164 and 184-185: The authors use the function "Enhance contrast" to automatically find the boundaries of the joining partners after binarizing the image using a threshold of 230. The authors could include the difference between the mean gray level of the image and the threshold, as this could help other researchers.

- Lines 220-222 and 238-243: Could the authors explain further why some weld seams could not be processed? why only 3rd to 6th B-scans are identified?

- The authors shouls indicate if the impossibility to process all the samples will be a disadvantage during quality control in industrial applications. I suppose the answer is no as usually quality control only requires the evaluation of some samples.

- Figure 4: This figure shows seam width is overstimated when using OCT. I suppose the processing parameters have been adjusted to obtain the best adjustment for U and L distances. Am I right?. If so, that information should be included in the text.

Reviewer 3 Report

Optical coherence tomography for 3D weld seam localization in absorber-free laser transmission welding

In this work, the authors reported the results of their experimental study on using OCT for 3D weld seam localization in absorber-free laser transmission welding. The work is well written and structured. However, the manuscript needs to be revised significantly based on the following comments:

Major corrections:

- It is hard to understand, what is the novelty of the work. The authors claim that “we employ a spectral domain OCT (SD-OCT)”, however, SD-OCT was already used in previous studies (e.g., https://doi.org/10.1109/ACCESS.2018.2882527, and some more works available in literature) for the same purpose (Identifying the geometry of welding area in Laser Transmission Welding). So, the authors need to clarify precisely the originality of this work? What makes it different from the previous ones.

- This point is complementary to the previous comment. The introduction chapter was presented appropriately. However, no relevant works were mentioned to show state of the art for the OCT (except one [11] in the last paragraph). So, I would recommend the authors briefly explain (in one short paragraph) the finding of at least three relevant previous studies and what is new in current work than those.

- Authors should be aware that when they use any Figure published previously (whether in their own publications or others) to cite that article. For example, the schematic diagram in Figure 2 was almost 90% taken from a previous article (with slight modifications) belonging to some of the co-authors (i.e., Fig. 2 in https://doi.org/10.1016/j.phpro.2014.08.055). This is considered plagiarism; even if the published paper was yours, it should be cited.

- Obviously, this work has no discussion as it is totally absent. Therefore, please change this chapter's title to "Conclusions", and create a new chapter to add some discussion, or you may combine the discussion with the Results chapter as it is also rather very short. In the discussion, you can give some justifications for the results obtained. Also, to extend the discussion a bit further, you may compare your results with the outcomes of the previous researches.

Minor corrections:

- Line 26: The abbreviations should be written when the abbreviated words are first mentioned. Therefore, OCT should be mentioned in line 24 first after "optical coherence tomography" directly, otherwise, it will be hard to recognize what does it mean if it comes the first time alone.

- Figure 1: Since you have already added "left" after the caption of the left part of the Figure, you should add the word "right" for the right part as well.

- Line 66: Either (i.e., gap, bubbles, etc.) or (i.e., gap and bubbles). The same for line 99, either (i.e., air, combustion gases, etc.) or (i.e., air, and combustion gases)

- Line 107: Please provide details about the brand and manufacturer (with its country) of the laser system employed.

- 118: Please identify the country of the provider of polyamide 6 (BASF 117 Ultramid B3S). Please do the same for all equipment and materials used in the Materials and Methods chapter.

- Figures 4 and 5: Why the first word of the x-axis and y-axis does not start with a capital letter? Also, the resolution of Figure 4 is low, please try to improve it.

Round 2

Reviewer 1 Report

The comments were addressed properly.

Reviewer 3 Report

Dear Authors,

Congratulations on your work. It seems that you have improved the manuscript significantly according to the recommendations. Therefore, I will recommend accepting this article for publication in Applied Science.

Best of luck.